# Evaluation of Ischemia with No Obstructive Coronary Arteries (INOCA) and Contemporary Applications of Cardiac Magnetic Resonance (CMR)

**DOI:** 10.3390/medicina59091570

**Published:** 2023-08-29

**Authors:** Andrew Chang, Nicolas Kang, Joseph Chung, Aakash Rai Gupta, Purvi Parwani

**Affiliations:** 1Division of Cardiology, Department of Medicine, Loma Linda University Health, Loma Linda, CA 92374, USA; akchang@llu.edu (A.C.); jjchung@llu.edu (J.C.); 2Department of Medicine, Loma Linda University Health, Loma Linda, CA 92374, USA; nkang@llu.edu (N.K.); agupta@llu.edu (A.R.G.)

**Keywords:** INOCA, CMR, ischemia with no obstructive coronary arteries, cardiac magnetic resonance, diagnosis

## Abstract

Ischemia with no obstructive coronary arteries (INOCA) is a relatively newly discovered ischemic phenotype that affects patients similarly to obstructive coronary artery disease (CAD) but has a unique pathophysiology and epidemiology. Patients with INOCA present with ischemic signs and symptoms but no obstructive CAD seen on coronary CTA or invasive coronary angiography, which can assess epicardial vessels. The mechanisms of INOCA can be grouped into three endotypes: coronary microvascular dysfunction, epicardial coronary vasospasm, or a combination of both. Accurate and comprehensive assessment of both epicardial and microvascular disease in suspected cases of INOCA is crucial for providing targeted therapy and improving outcomes in this underrepresented population. This review aims to clarify the complex pathophysiology of INOCA, present an overview of invasive and non-invasive diagnostic methods, and examine contemporary approaches for coronary perfusion assessment using cardiac magnetic resonance (CMR). We also explore how recent advancements in quantitative CMR can potentially revolutionize the evaluation of suspected INOCA by offering a rapid, accurate, and non-invasive diagnostic approach, thereby reducing the alarming number of cases that go undetected.

## 1. Introduction

First discovered in 1973 by Harvey Kemp [1], ischemia with no obstructive coronary arteries (INOCA) refers to patients with stable ischemic symptoms and visually normal or non-obstructive coronary arteries (i.e., <50% reduction in coronary artery diameter appreciated on invasive or CT angiography) [2]. The mechanism of INOCA involves coronary microvascular dysfunction and/or coronary vasospasm, which is explained further below. Though initially considered benign, INOCA is now known to represent true cardiac disease and increases morbidity and mortality. For accurate diagnosis and risk stratification of these patients, a combination of anatomical and functional testing of the epicardial arteries and the microvasculature is crucial [3]. However, diagnostic tests remain underutilized in INOCA [3]; as the current gold standard, coronary reactivity testing is invasive, expensive, and technically challenging [4,5]. Fortunately, non-invasive perfusion imaging techniques are emerging that can provide comparably accurate vasomotor measurements and lower the barrier for evaluation [3,6,7].

This review discusses the endotypes of INOCA, explores its underlying pathophysiology, offers a concise overview of the existing evaluation methods for INOCA, and emphasizes contemporary approaches for myocardial perfusion analysis utilizing cardiac magnetic resonance (CMR).

## 2. Epidemiology, Prevalence, Risk Factors, and Outcomes of INOCA

INOCA is a relatively common condition, affecting 3 to 4 million individuals in the United States alone [8]. Among patients undergoing coronary angiography for suspected angina, approximately 60% of women and 30% of men have INOCA [8]. It is thought that this demographic disparity in the prevalence of INOCA is due to sex-related differences in normal cardiac physiology as well as a heterogenous representation of risk factors in women compared to men [8]. There are several risk factors associated with INOCA, including traditional cardiovascular risk factors (i.e., hypertension, diabetes, hyperlipidemia, age, and smoking), non-traditional risk factors (i.e., psychosocial stress, autoimmune disorders, and hormonal changes), female sex, and postmenopausal status [9,10]. There is also growing evidence for drug-induced INOCA, including one study that found microvascular disease associated with the use of anthracycline [11].

Several studies have highlighted the significant morbidity and mortality associated with INOCA, including an increased risk of major adverse cardiac events (MACE), heart failure with preserved ejection fraction (HFpEF), stroke, and coronary microvascular dysfunction (CMD) [9,10]. INOCA has been found to increase the risk of MACE 1.5–1.8-fold and the risk of all-cause mortality 1.3–1.5-fold [9,10,12]. The WISE study demonstrated that symptomatic women with INOCA experienced a 10-fold increase in heart failure hospitalizations compared to healthy asymptomatic women [10]. Overall, INOCA reduces a patient’s quality of life with increased symptom burden, cardiac anxiety, emergency room visits, and repeated testing with invasive angiography [13,14]. INOCA also comprises a substantial healthcare burden, with costs comparable to obstructive CAD [9]. It accounts for almost half of all angiography procedures [10,15].

Unfortunately, INOCA is often under-detected and undertreated in both men and women due to limitations in our current diagnostic tools, inadequate awareness, and bias [3]. Early identification and appropriate management of INOCA are essential to reduce the risk of adverse outcomes in affected individuals.

## 3. INOCA Endotypes: Pathophysiology and Current Diagnostic Criteria

The coronary microvasculature (especially the small arterioles) provides a significant component of the overall coronary vascular resistance and is thereby a pivotal regulator of myocardial blood flow [16]. Disturbances in the microvascular structure and/or vasodilator responses can lead to INOCA [16].

The two endotypes of INOCA include microvascular dysfunction (MVD) and vasospastic angina (VSA). They represent distinct but frequently coexistent mechanisms [17]. A meta-analysis examining the distribution of these endotypes found that approximately 41% of INOCA cases are MVD, 40% are VSA, and 23% are a combined endotype [3]. Table 1 outlines the diagnostic criteria for each endotype and compares them to non-cardiac chest pain [18].

Physiologically, endothelium-independent and endothelium-dependent vasodilator responses regulate the coronary vasculature [16]. Adenosine, which is released by cardiac tissue in the setting of insufficient oxygen supply, stimulates endothelium-independent vasodilation in the microvasculature in order to increase myocardial blood flow (MBF) [16]. This process then promotes flow-mediated acetylcholine (ACh) release, which triggers the endothelium to produce nitric oxide (NO), causing endothelium-dependent vasodilation in larger epicardial and microvascular vessels [16,17]. ACh also induces smooth muscle vasoconstriction, but normally, endothelium-dependent vasodilation predominates [19,20]. Additionally, other chemical stimuli like histamine, bradykinin, serotonin, adenosine diphosphate (ADP), substance P, and thrombin can trigger NO release, contributing to further vasodilation [16].

Aberrancies in the above pathway can lead to INOCA [16]. Just as atherosclerotic vascular disease is known to involve remodeling of the arteries, the coronary microvasculature can develop structural changes that can impair vasodilation and flow. The coronary arteries may also develop functional dysregulation, leading to enhanced response to vasoconstrictive stimuli. When the endothelium fails to release NO in response to ACh, ACh will induce unrestrained smooth muscle vasoconstriction, leading to vasospasm [19]. The physiology of coronary vasodilation and pathophysiology of INOCA are illustrated in Figure 1.

### 3.1. Microvascular Dysfunction Endotype (MVD)

The MVD endotype is characterized by coronary microvascular disease due to structural remodeling and/or microvascular vasospasm. Structural remodeling in the microvasculature leads to an increased wall-to-lumen ratio and a loss of myocardial capillary density (capillary rarefaction) [17]. This increases the index of microvascular resistance (IMR) and impairs vasodilation [17]. Traditional cardiovascular risk factors (i.e., smoking, hypertension, hyperlipidemia, diabetes, insulin resistance, and obstructive CAD), left ventricular hypertrophy, and cardiomyopathies can predispose one to structural microvascular remodeling, which can decrease microvascular vasodilatory capacity and limit the blood and oxygen reserve to the myocardium in response to stress or exercise [17]. Microvascular vasospasm due to enhanced vasoreactivity may also comprise part of MVD’s pathophysiology.

The current criteria for diagnosing MVD include the clinical presence of myocardial ischemia (i.e., symptoms of angina and/or ECG changes during stress testing), the absence of obstructive CAD on coronary CTA or invasive angiography, and evidence of microvascular dysfunction on invasive or non-invasive coronary reactivity testing [4]. The findings of coronary microvascular dysfunction include a measurement of CFR ≤ 2 or 2.5 after adenosine administration, an IMR > 25, and/or a corrected TIMI frame count ≥ 3 beats to fill a vessel [4,21,22].

### 3.2. Vasospastic Angina Endotype (VSA)

As opposed to the microvascular disease found in MVD, VSA is characterized by epicardial dysfunction without evidence of obstruction. Though functional dysregulation may lead to enhanced vasoconstriction in either epicardial and/or microvascular vessels [17], VSA is defined by coronary vasospasm that is predominately evidenced in the epicardial vessels [19,20,23].

According to the COVADIS criteria, definitive vasospastic angina is diagnosed when nitrate-responsive angina (either during a spontaneous episode or due to a trigger such as exercise, hyperventilation, or acetylcholine stimulation) is accompanied by either transient ischemic electrocardiographic changes (e.g., ST elevation or depression ≥ 0.1 mV, new negative U-wave) or coronary artery spasm (e.g., >90% coronary artery constriction) [5].

## 4. Invasive Coronary Reactivity Testing (CRT)

CRT, the current gold standard for diagnosing INOCA, measures coronary blood flow (CBF), coronary flow reserve (CFR), thrombolysis in myocardial infarction (TIMI) flow count, and index of microvascular resistance (IMR) [4,24]. Decreased coronary blood flow can be identified by a corrected TIMI flow count that demonstrates the slow flow phenomenon or angiography that shows delayed contrast filling in a vessel [22]. CFR, measured by intracoronary doppler or thermodilution, assesses endothelium-independent vasodilation [25]. A reduced CFR indicates an impaired vasodilatory response to meet metabolic demands during hyperemic states [25]. The diagnosis of coronary microvascular dysfunction requires either a low CFR, a high IMR, or a high TIMI frame count [4,21,22].

To assess endothelial dysfunction that leads to vasospasm at the epicardial and/or microvascular level, one can perform invasive provocative testing using intracoronary acetylcholine, which induces coronary vasoconstriction and reproduces ischemic symptoms [2]. Vasospasm is defined as greater than 90% constriction in the coronary artery following provocative testing [5].

CRT is invasive, expensive, not typically covered by insurance, and not readily available, prompting this investigation into more viable methods for diagnosis.

## 5. Non-Invasive Methods for Evaluating INOCA

There is growing interest in the use of non-invasive imaging techniques for evaluating INOCA, with CMR being the most promising due to its high spatial resolution and tissue characterization without radiation exposure. Other modalities such as quantitative perfusion, cardiac PET, and CT scans can also play a role. These imaging methods typically involve the administration of a “stressor” or vasodilator, such as adenosine or regadenoson. Contraindications to the use of a vasodilator include the presence of significant reactive airway disease, advanced conduction abnormalities in the absence of a pacemaker, unstable blood pressure, recent caffeine intake, or recent acute coronary syndrome [25,26].

### 5.1. Coronary CT Angiography (CTA)

Coronary CTA (CCTA) can evaluate myocardial perfusion by tracking the movement of contrast from the coronary vasculature into myocardial tissue during rest and after adenosine administration [5]. Contrast attenuation occurs when there is reduced myocardial perfusion, as less contrast agent enters myocardial tissue [5]. The proprietary software HeartFlow can quantify fractional flow reserve (FFRCT) at a specific region in the coronary tree by simulating maximum hyperemia. However, iodinated contrast itself may stimulate coronary vasodilation and, therefore, overestimate coronary blood flow [27]. The diagnostic utility of cardiac CT for coronary blood flow assessment should be balanced with the risk of high radiation exposure and contrast nephropathy in patients with chronic kidney disease [27]. CCTA, however, is unable to measure CFR and FFRCT provides flow information about the limited vasculature.

### 5.2. Cardiac Positron Emission Tomography (PET)

Cardiac PET is a well-validated imaging modality that can measure myocardial blood flow (MBF), myocardial perfusion reserve (MPR), and myocardial flow reserve (MFR, or the ratio of MBF during hyperemia and at rest) [5]. PET measures MBF using radiotracers (i.e., rubidium-82 and nitrogen-13 ammonia), which are detected as they diffuse across the myocardial cell membrane into the cell [28]. Given that cardiac PET can assess the degree of vascular dysfunction, it may be useful for risk assessment; for example, reduced MBF has been associated with greater risk for diastolic dysfunction and HFpEF-related hospitalizations [28]. Significant barriers to routine use include high costs, radiation exposure, and inadequate availability. The level of radiation exposure is generally considered safe but does pose increased risk to pregnant or reproductive-aged women, which is a large fraction of the population affected by INOCA [28].

### 5.3. Cardiac Magnetic Resonance (CMR)

CMR is a powerful, non-invasive imaging modality that provides comprehensive evaluation of myocardial structure, function, and both global and regional perfusion without the exposure to ionizing radiation [29]. CMR is commonly used to evaluate LV morphology, contractility, viability, and ischemia in patients with chronic ischemic heart disease [30]. Follow-up imaging can be used to assess changes in ventricular remodeling, scarring, and/or ischemia burden [30]. With the added capabilities of stress CMR and quantitative perfusion techniques, one can gain insights into myocardial ischemia and microvascular function that rival the gold standard, invasive CRT [29].

## 6. Contemporary Applications of CMR in INOCA Evaluation

Though CMR is a relatively new modality, it has been shown to provide an unmatched breadth of qualitative and quantitative information on cardiac structure and function. Different techniques and analyzing software are emerging that each examine the heart from a unique lens. Here, we compare several applications of CMR in their ability to assess INOCA. The proposed diagnostic algorithm of INOCA, including the use of CMR, is illustrated in Figure 2.

### 6.1. First-Pass Sequence Perfusion Imaging

When using stress CMR, first-pass sequence perfusion imaging is the most common approach to evaluate coronary perfusion [31]. Vasodilators (i.e., adenosine, regadenoson, or dipyridamole) are administered to induce hyperemia, after which a gadolinium-based contrast agent is injected to serve as a blood flow tracer [31]. Serial T1-weighted CMR images are then taken during each heartbeat to visualize the contrast agent perfusing the myocardium [31]. Higher contrast penetration into well-perfused tissue increases the T1 signal, appearing brighter than poorly perfused myocardium and preferentially involving the subendocardial layers [31]. Visual assessment of an attenuated signal in a myocardial region, in the absence of obstructive CAD, may represent impaired myocardial blood flow and microvascular disease. However, this visual assessment is a qualitative measure of perfusion and may be inaccurate when MBF is globally reduced, as in three-vessel CAD [32]. In fact, some studies have found that qualitative visual analysis using CMR is only able to detect significant multi-vessel CAD in up to two-thirds of patients [33,34]. Furthermore, Rahman et al. [35] showed that visual assessment of poorly perfused myocardial segments had poor accuracy in detecting coronary microvascular disease when compared to invasive CRT measurements (diagnostic accuracy: 58%, specificity: 41%, sensitivity: 83%), as seen in Table 2.

### 6.2. Semi-Quantitative CMR

Semi-quantitative methods of measuring myocardial perfusion use a parameter called the up-slope with a correction factor known as the arterial input function (AIF). The up-slope represents the signal intensity in the myocardium enhancing over time as contrast is injected. The up-slope has an approximately linear dependence on blood flow [36]. The correction factor, AIF, is required because the up-slope can be influenced by hemodynamic conditions [36], and it is calculated by measuring the up-slope of the signal intensity in the blood in the left ventricle [37]. The ratio of the up-slope during stress to the up-slope at rest can estimate the myocardial perfusion reserve (MPR) [36]. Studies have shown that semi-quantitative perfusion CMR could correctly identify patients who have CRT-proven coronary microvascular disease [38]. Thomson et al. [38] showed that at an MPR cutoff of ≤1.84, semi-quantitative CMR predicted at least one CRT abnormality with reasonable diagnostic accuracy (AUC: 78%, sensitivity: 73%, specificity: 74%), as seen in Table 2. However, while semi-quantitative CMR is more accurate than visual assessment alone, it comes at the cost of significant time penalties when processing perfusion data [39].

### 6.3. Quantitative CMR: Dual-Bolus and Dual-Sequence Methods

Quantitative perfusion methods involve precise measurements of myocardial enhancement, requiring advanced computational power [39]. During the first pass, gadolinium concentration is higher in the blood pool than in the myocardium, leading to T1, T2, and T2* signal variations not seen when the gadolinium is diluted [39]. To mitigate this issue, dual-bolus or dual-sequence approaches can be utilized [39]. The dual-bolus method involves performing perfusion scans at rest and during stress with an initial low-dose gadolinium injection, followed by another scan with a standard gadolinium dose [39]. In the dual-sequence approach, two distinct sequences are used—a full-coverage sequence optimized for the myocardium and a single-slice sequence specifically designed to measure blood with high gadolinium concentrations, enabling more accurate AIF measurements [39].

Rahman et al. [35] reported in an observational study that quantitative stress and rest perfusion CMR using a dual-bolus strategy outperformed visual assessment and stress-only protocols in detecting microvascular dysfunction (MVD, defined as CFR < 2.5). Specifically, subendocardial MPR with a cutoff ≤2.41 demonstrated an AUC of 0.90 for identifying MVD, whereas visual analysis achieved an AUC of 0.60 and quantitative stress MBF alone achieved a similar AUC of 0.64. Results of the study are seen in Table 2. Quantitative CMR also results in lower interobserver variability compared to visual assessment [39].

Developing a fully quantitative CMR technique has the benefit of improved diagnostic accuracy but also the ability to characterize disease severity [39]. Studies found that, unlike visual assessment, quantitative CMR can differentiate between one-vessel, three-vessel, and microvascular involvement [32,39]. Another study found that MPR can provide prognostic value—when the MPR decreased by one unit, the adjusted hazard ratio (HR) increased by 2.22 for death and 1.65 for MACE [40].

These quantitative methods require more complex imaging protocols and may be more time-consuming, and so they would benefit from a fully automated approach [39].

### 6.4. Automated In-Line CMR Perfusion Mapping

In-line CMR uses a novel respiratory motion-corrected myocardial perfusion method with automated perfusion mapping, allowing free-breathing acquisition and pixel-wise quantification of MBF [32]. In-line CMR measures pulse sequences for both AIF and myocardial tissue but then achieves linearity between signal and contrast agent concentration by automatically incorporating several correction factors, including surface coil sensitivity and T2* losses [41]. This allows the user to objectively assess the severity of CAD, diagnose MVD, and risk-stratify patients in minutes [32]. It also provides more accurate measurements of MBF when global MBF is reduced, which helps to differentiate between multivessel CAD and MVD [32].

Kotecha et al. [32] examined the performance of regional stress MBF, MPR, and global MBF in predicting increased IMR and differentiating between types of CAD. They demonstrated that regional stress MBF ≤ 2.19 mL/g/min effectively predicted an IMR ≥ 25, with an AUC of 0.73, sensitivity of 71%, and specificity of 70%. An MPR cutoff of ≤2.06 also predicted increased IMR with similar accuracy. The study also showed that both regional stress MBF ≤ 1.94 and global MBF ≤ 1.84 accurately detected obstructive CAD and differentiated three-vessel CAD from MVD with high AUC and statistical significance (*p*-value = 0.001).

## 7. Conclusions

The scientific community has made great strides in developing the understanding of the clinical significance of INOCA and its disproportionate effect on women. The next challenge is diagnosing INOCA in a manner that is cost-effective, readily accessible, and less invasive than coronary reactivity testing. By applying creative techniques to well-established cardiac imaging modalities, we can lower the barriers for INOCA testing and reduce our high rates of underdiagnosis. CMR perfusion imaging seems to be the most promising, where novel automated software can quickly and non-invasively provide diagnostic data that rival the gold standard.

## 8. Future Directions

Given the high burden of INOCA and the difficulties in detection, there is growing interest in optimizing our methods for diagnosis. Preliminary studies have demonstrated the potential for non-invasive cardiac imaging in characterizing INOCA, but more work is needed to confirm their accuracy and to learn how they can provide more refined data to stratify patients by disease severity and/or subtype. With more accessible and more revealing diagnostic techniques, we can reduce the number of patients with INOCA who are suffering silently and connect them to the care they need.

## Figures and Tables

**Figure 1 medicina-59-01570-f001:**
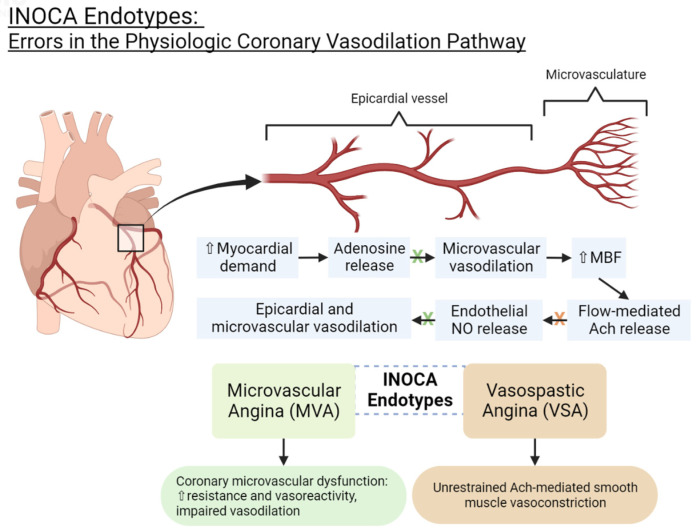
INOCA endotypes: errors in the physiologic coronary vasodilation pathway. NO = nitric oxide. Ach = acetylcholine. MBF = myocardial blood flow.

**Figure 2 medicina-59-01570-f002:**
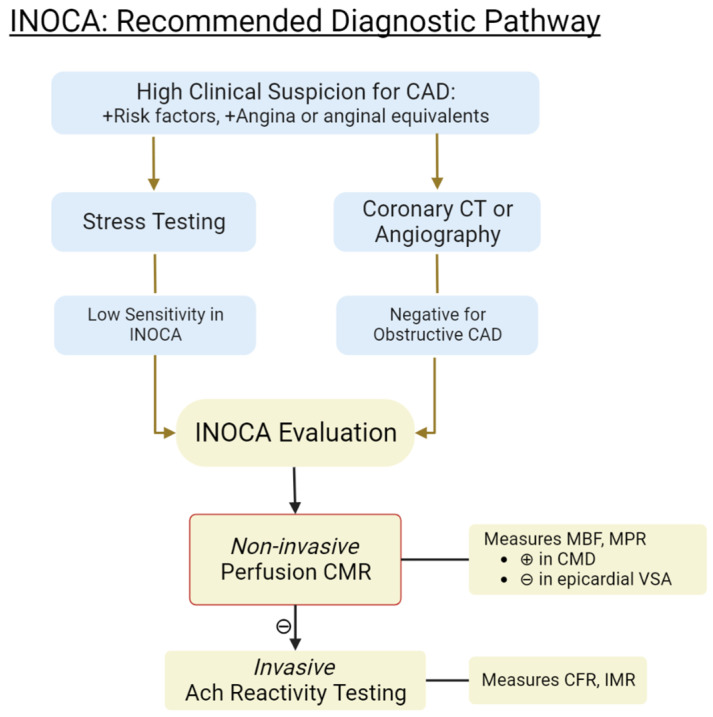
Recommended diagnostic pathway for INOCA. CAD = coronary artery disease. CMR = cardiac magnetic resonance. MBF = myocardial blood flow. MPR = myocardial perfusion reserve. CMD = coronary microvascular dysfunction. VSA = vasospastic angina. CFR = coronary flow reserve. IMR = index of microvascular resistance. Ach = acetylcholine.

**Table 1 medicina-59-01570-t001:** Diagnostic criteria for INOCA endotypes and non-cardiac chest pain.

Diagnosis	Diagnostic Criteria
Microvascular dysfunction (MVD) *	Symptoms of myocardial ischemia: Exertional and/or rest angina; Anginal equivalents (i.e., dyspnea, arm pain). Absence of obstructive CAD by CTA or invasive angiography: Diameter reduction >50%; FFR < 0.80. Objective signs of myocardial ischemia: Ischemic ECG changes during an episode of chest pain; Stress-induced chest pain or ischemic ECG changes in the presence or absence of transient/reversible abnormal myocardial perfusion and/or wall motion abnormality. Objective signs of microvascular dysfunction: Impaired coronary flow reserve (CFR ≤ 2.0 and ≤ 2.5 depending on the methodology); Abnormal index of coronary microvascular resistance (IMR > 25); Coronary slow flow phenomenon (TIMI frame count > 25); Coronary microvascular spasm (i.e., ischemic symptoms or ECG changes during acetylcholine testing without epicardial spasm).
Vasospastic angina (VSA) **	A spontaneous episode of nitrate-responsive angina with at least one of the following: Rest angina; Marked diurnal variation in exercise tolerance; Hyperventilation-induced episodes; Relief with calcium channel blockers (not b-blockers). A spontaneous episode of transient ischemic ECG changes; A spontaneous or provoked epicardial coronary artery spasm with angina and ischemic ECG changes.
Non-cardiac chest pain	Absence of epicardial coronary disease: Coronary artery diameter reduction of <50%; FFR > 0.80. Normal coronary vascular function: CFR > 2.0 or 2.5, depending on the study; IMR < 25; Absence of vasospasm following ACh testing.

* Microvascular dysfunction is definitively diagnosed only when all four criteria are met. If there are symptoms of ischemia but no obstructive coronary artery disease, suspected MVD may be diagnosed based on either objective evidence of myocardial ischemia or evidence of impaired coronary microvascular function alone. ** Definitive vasospastic angina is diagnosed when nitrate-responsive angina occurs during spontaneous episodes, along with either transient ischemic ECG changes or coronary artery spasm. Suspected vasospastic angina is diagnosed when nitrate-responsive angina occurs during spontaneous episodes, but transient ischemic ECG changes or coronary artery spasm criteria are uncertain or unavailable. Diagnostic criteria were established by the Coronary Vasomotion Disorders International Study Group (COVADIS) [4,5].

**Table 2 medicina-59-01570-t002:** Diagnostic utility of coronary perfusion methods using cardiac magnetic resonance (CMR).

Protocol	Assessment	Reference Standard	AUC (95% CI)	Sensitivity (95% CI)	Specificity (95% CI)
First-pass perfusion qualitative assessment	Visual assessment	CFR < 2.5	0.60	58%	83%
			(0.46–0.69)	(46–69%)	(65–94%)
Semiquantitative CMR	MPR < 1.84	≥1 CRT abnormality	0.78	73%	74%
			(0.68–0.88)	(64–82%)	(58–90%)
Dual-bolus fully quantitative perfusion CMR	MPR_ENDO_ ≤ 2.41	CFR < 2.5	0.90	95%	72%
			(0.82–0.97)	(83–99%)	(52–87%)
	MPR ≤ 2.19	CFR < 2.5	0.88	70%	90%
			(0.78–0.96)	(53–83%)	(74–98%)
Automated in-line CMR perfusion mapping	Regional stress MBF ≤ 2.19	IMR ≥ 25	0.73	71%	70%
			(0.63–0.84)		
	Regional MPR ≤ 2.06	IMR ≥ 25	0.68	44%	92%
			(0.56–0.80)		

Abbreviations: AUC = area under the curve; CMR = cardiac magnetic resonance; CRT = coronary reactivity testing; IMR = index of microcirculatory resistance; MPRENDO = subendocardial myocardial perfusion reserve; MPR = myocardial perfusion reserve; MBF = myocardial blood flow.

## Data Availability

No new data were created or analyzed in this study.

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
