# Peer review of "Evaluation of Ischemia with No Obstructive Coronary Arteries (INOCA) and Contemporary Applications of Cardiac Magnetic Resonance (CMR)"

_medicina, 2023, doi:10.3390/medicina59091570_

Round 1

Reviewer 1 Report

Lines 289-303: Why do we have these guidelines in the main manuscript?
Lines 312-370: Must be filled accordingly

Author Response

Lines 289-303: Why do we have these guidelines in the main manuscript?

Response: This was not part of the manuscript I submitted, I will delete.Lines 312-370: Must be filled accordingly

Response: These sections have been updated.

Reviewer 2 Report

The article addresses a very important, growing and under-reported problem of microcirculatory disorders. 

+ A detailed review of invasive and non-invasive studies of microcirculatory assessment

+ highlighted the growing role of CMR

Contraindications to the use of adenosine and regadenoson are lacking.

Microcirculatory disorders can also be caused by drugs, including chemotherapy (doxorubicin). This is a growing population of people after cancer treatment with disorders of the microcirculation as well. Indicated citation in the paper: 

Anthracycline-induced microcirculation disorders: the AIM PILOT Study.Klotzka A, Iwańczyk S, Ropacka-Lesiak M, Misan N, Lesiak M.Kardiol Pol. 2023 May 16. doi: 10.33963/KP.a2023.0108

Author Response

Contraindications to the use of adenosine and regadenoson are lacking.

Response: added contraindications to vasodilator administration to the section titled "Non-invasive Methods for Evaluating INOCA".

Microcirculatory disorders can also be caused by drugs, including chemotherapy (doxorubicin). This is a growing population of people after cancer treatment with disorders of the microcirculation as well. Indicated citation in the paper: 

Anthracycline-induced microcirculation disorders: the AIM PILOT Study.Klotzka A, Iwańczyk S, Ropacka-Lesiak M, Misan N, Lesiak M.Kardiol Pol. 2023 May 16. doi: 10.33963/KP.a2023.0108

Response: added a statement on this and your recommended citation to the section titled "Epidemiology, Prevalence, Risk Factors, and Outcomes of INOCA".

Reviewer 3 Report

The paper reports an overview of the INOCA diagnostic tool and the CMR's potential role, which could be of rising importance in consideration of the actual gold standard methodology represented by the invasive coronary reactivity test. The physiopathology and different subtypes of INOCA are adequately defined in the first paragraphs; nevertheless, there are some observations about the other sections:

- Table 1 displays the diagnostic criteria for each INOCA category, but the bibliographic source from which they were watched is not shown. Please, add guidelines and/or statements about it.

- in the "Semi-quantitative CMR" paragraph, the abbreviation for myocardial perfusion reserve index is reported as MPRI. In the following sections was displayed as "MPR". The choice between either should avoid confusion. 

- in the "Automated In-line CMR Perfusion mapping" paragraph, the main steps regarding the methodology are not detailed, different from the other methods described in the previous paragraphs (semiquantitative and quantitative CMR).

- Table 2 shows the diagnostic utility of CMR protocols. The dual-bolus quantitative and automated perfusion mapping appear twice because of different assessment criteria used in the studies. This presentation should represent a confounding manner to present the data. CMR perfusion evaluation, as reported in the paper, has qualitative, semi-quantitative, and quantitative methods. A better way to outline the different assessment protocols should be to include two different subcategories within the same protocol in the table. 

- About Conclusions, the authors underline the underrepresentation of women in the INOCA studies. Although this could be a reasonable conclusion, the concept is not deeply analyzed in the text, because there are no specific data from the studies reported for gender within the text.  

Author Response

-Table 1 displays the diagnostic criteria for each INOCA category, but the bibliographic source from which they were watched is not shown. Please, add guidelines and/or statements about it.

Response: Added statement on source into Table 1

-In the "Semi-quantitative CMR" paragraph, the abbreviation for myocardial perfusion reserve index is reported as MPRI. In the following sections was displayed as "MPR". The choice between either should avoid confusion. 

Response: changed all "MPRI" to "MPR" for consistency

-In the "Automated In-line CMR Perfusion mapping" paragraph, the main steps regarding the methodology are not detailed, different from the other methods described in the previous paragraphs (semiquantitative and quantitative CMR).

Response: added in more details on the methodology into this section.

-Table 2 shows the diagnostic utility of CMR protocols. The dual-bolus quantitative and automated perfusion mapping appear twice because of different assessment criteria used in the studies. This presentation should represent a confounding manner to present the data. CMR perfusion evaluation, as reported in the paper, has qualitative, semi-quantitative, and quantitative methods. A better way to outline the different assessment protocols should be to include two different subcategories within the same protocol in the table. 

Response: edited Table 2 as you recommended, with subcategories, instead of having the same protocol appear twice.

-About Conclusions, the authors underline the underrepresentation of women in the INOCA studies. Although this could be a reasonable conclusion, the concept is not deeply analyzed in the text, because there are no specific data from the studies reported for gender within the text.

Response: statement on underrepresentation of women in the INOCA studies has been deleted. Kept the statement on how INOCA affects women more than men, which is discussed in detail in the main text.